# OpenReview forum: "DecodingTrust: A Comprehensive Assessment of Trustworthiness in GPT Models"
_NeurIPS.cc/2023/Track/Datasets_and_Benchmarks — NeurIPS 2023 Datasets and Benchmarks Oral_

### Official Review · Reviewer_imyr · 2023-07-10
**Trustworthiness in GPT Models**

**Rating:** 7
**Confidence:** 4
**Clarity:** The paper is well-structured and clear.

**Strengths:**

1） The paper has comprehensive analysis of trustworthiness across multiple dimensions. This holistic approach gives us a better understanding of the LLMs’ performance and their potential vulnerabilities.

2） This paper finds that GPT-4 is easier to manipulate due to its higher instruction-following capability, telling the community that we should also consider trustworthiness alongside performance in LLMs.



**Additional Feedback:**

NA

**Correctness:**

The methodology appears to be sound. The paper clearly describes the metrics used to evaluate the models and the design of evaluations.

**Documentation:**

Yes, a public github link is provided.

**Ethics:**

Please see limitation section.

**Limitations:**

When the paper shares examples of unsafe or private prompts, it could give people with bad intentions ideas on how to misuse these models. It might be a good idea for them to keep these specific prompts private and only share them if someone asks for them for research purposes. Balancing between being open in their research and avoiding potential misuse of the information is tricky, but it's something important the authors need to tackle.

Also, one question that arises from the paper's approach is the reliance on specific prompts to trigger trustworthiness issues in the LLMs. If the developers of these models were to correct the problems highlighted by these prompts, the revised models might appear more trustworthy and robust when re-evaluated using the same prompts. However, this may not necessarily mean the models have become 'truly' more trustworthy in a broader sense. It could just mean they have been optimized to pass these specific tests. Should the keep a private set of prompts to evaluate to ensure more robust testing? Or could there be an automated solution that continually generates new prompts to challenge the models?


**Opportunities For Improvement:**

It could benefit from a more explicit discussion of potential solutions or strategies to address the identified vulnerabilities in the LLMs.

**Relation To Prior Work:**

Yes

**Summary And Contributions:**

This paper provides a comprehensive evaluation of the trustworthiness of two large language models (LLMs), GPT-3.5 and GPT-4, across a range of aspects including toxicity, stereotype bias, adversarial robustness, out-of-distribution robustness, adversarial demonstrations, privacy, machine ethics, and fairness. Its novelty lies in the thorough comparison of these two models and the identification of trustworthiness factors that affect the performance of these LLMs. This paper reveals that although GPT4 and GPT3.5 achieve superior performance on natural language benchmarks, they can be easily manipulated by adversarial attacks or hand-crafted toxic prompts.

This kind of paper is definitely needed in the NLP and LLM community. It's so important to get a clear understanding of how trustworthy these big language models are, and this paper does a great job of diving into that.

---

> ### Author Response · Authors · 2023-08-22
> **Thank you for your valuable comments**
>
> We thank the reviewer for the valuable comments and feedbacks. We are glad that the reviewer finds our work conducting comprehensive analysis, providing holistic approach, giving a better understanding of the LLM’s performance and their potential vulnerabilities, and doing a great job diving into LLM understandings. We provide details response as below.
>
> 1. “It could benefit from a more explicit discussion of potential solutions or strategies to address the identified vulnerabilities in the LLMs.”
>
> - Thanks for the insightful suggestion. We have listed several potential solutions and strategies to mitigate potential vulnerabilities in the LLMs in Appendix M. Specifically, we mention four potential ways to safeguard the trustworthiness of LLMs, including:
>
>     -  **Safeguarding LLMs with additional knowledge and reasoning analysis**: PAs purely data-driven models such as GPT models would suffer from the imperfection of the training data and lack of reasoning capabilities in various tasks. Thus, it would be important to equip domain knowledge and logical reasoning capabilities for language models and safeguard their outputs to make sure they satisfy basic, domain knowledge and logic to ensure the trustworthiness of the model outputs.
>
>     - **Safeguarding LLMs based on consistency checking**: Our designed system prompts based on “role-playing" shows that models can be easily fooled based on role-changing and manipulation. This indicates that during the conversation of LLMs, it is possible to design diverse roles to ensure the consistency of the model’s answers, and therefore at least avoid the models being self-conflict. It is also possible to design different roles for the models to make sure it understands the context well to provide more informative and trustworthy answers.
>
>    - **Safeguarding LLMs via trustworthy finetuning**: Our generated challenging and adversarial prompts usually represent the long-tailed and “rare” events of the original training data distribution. As a result, it would be helpful to use our generated challenging prompts to finetune the LLMs and improve their trustworthiness. On the other hand, we note that new adaptive adversarial attacks could still be conducted against the new finetuned LLMs, and therefore we need to be aware of new adaptive attacks and try to provide certain trustworthiness verifications which are agnostic to actual attack strategies.
>
>     - **Verification for the trustworthiness of LLMs**: Empirical evaluations of LLMs are important but lack of guarantees, especially in safety-critical domains, so rigorous trustworthiness guarantees would be critical. To safeguard the trustworthiness of LLMs, it is important to provide verification for the trustworthiness of LLMs based on specific functionalities or properties, or map the discrete input space to their corresponding continuous space such as the embedding space with semantic preservation to perform verification leveraging existing verification tools in the continuous space.
>
> - We have incorporated the above discussions in Section 10 of our main text in the revision, and thank you for the valuable suggestions!

---

> ### Author Response · Authors · 2023-08-22
> **Thank you for your valuable comments**
>
> 2. “It might be a good idea for them to keep these specific prompts private and only share them if someone asks for them for research purposes. Balancing between being open in their research and avoiding potential misuse of the information is tricky, but it's something important the authors need to tackle… / Should they keep a private set of prompts to evaluate to ensure more robust testing? Or could there be an automated solution that continually generates new prompts to challenge the models?”
>
> - Thanks for the insightful comments. We agree that sharing of the adversarial experimental settings in this paper such as the jailbreaking prompts could be exploited by malicious users to misuse these models. Hence, it is important for us to balance between research openness and avoiding misuse of information. We have added more discussion about the potential social impacts in Appendix N in the revision.
>
> - First, we believe sharing the high-level settings of our evaluation will be beneficial for both researchers and practitioners who aim to train LLMs and understand the model capabilities, and need to be aware of the model vulnerabilities before deployment. In particular, our platform and findings provide comprehensive evaluations to understand the model capabilities and vulnerabilities, which is critical before deploying LLMs in practice. Similar to several concurrent efforts in exploring the vulnerabilities of LLMs [1,2,3], we aim to provide better understandings and insights about the models in adversarial environments, so that users could avoid the potential attacks.
>
> - In addition, we note that **our studies are able to generate more prompts automatically for all the perspectives**. We will share the prompts used in our evaluation while preserving some newly generated prompts for future evaluation to balance the research openness and information misuse. Taking the toxicity perspective as an example, the existing toxic sentences could be served as seed prompts for LLMs to generate coherent continuations which are later served as new challenging user prompts and jailbreaking prompts. Similarly, we can automatically generate more adversarial instances for AdvGLUE++ to test the adversarial robustness of LLMs, generate more testing prompts based on existing tabular datasets for fairness evaluation, and generate more privacy extraction prompts, etc. In principle, we will keep parts of the newly generated challenging data private to ensure that as a white-hat model evaluation, we are slightly ahead of the actual adversaries in the real world, so that we can start to design potential solutions against these vulnerabilities before they are implemented in practice.
>
> [1] Qiu, H., Zhang, S., Li, A., He, H., & Lan, Z. (2023). Latent Jailbreak: A Benchmark for Evaluating Text Safety and Output Robustness of Large Language Models. ArXiv, abs/2307.08487.
>
> [2] Liu, Y., Yao, Y., Ton, J., Zhang, X., Cheng, R.G., Klochkov, Y., Taufiq, M.F., & Li, H. (2023). Trustworthy LLMs: a Survey and Guideline for Evaluating Large Language Models' Alignment.
>
> [3] https://www.jailbreakchat.com/

---

### Official Review · Reviewer_qCen · 2023-07-21
**Thorough contribution with major concerns about the narrative and the level of details in the main text**

**Rating:** 7
**Confidence:** 4
**Clarity:** The paper is well-written and clear.

**Strengths:**

The contribution of this paper is a comprehensive assessment of the trustworthiness of GPT models, analysing a wide range of critical aspects. The authors present a thorough analysis of their benchmark methodology, designed to test the trustworthiness of LLMs, and apply their approach to two popular large language models that are already embedded in numerous products and services. The relevance of this study is heightened by the current context, where the safety and ethical implications of language models are currently under scrutiny.

The authors'  examination of various aspects of the GPT models is a crucial strength of the paper, making their findings valuable to the scientific community and beyond. Their approach and methodology are expected to be readily applied to other language models.

**Additional Feedback:**

While I have reservations about the contribution, I must concede that the paper  presents valuable insights and that the benchmark could be of interest to the community and beyond. Nonetheless, the concept of trustworthiness, which is a central theme throughout the paper, could benefit from greater emphasis and elaboration. In this regard, I feel that the narrative should be strengthened by expanding the discussion and introducing more unified views on the different persperctives, and/or thorough evaluation of different models beyond GPT models. Moreover, the current evaluation of the proposed framework is complex and presents several challenges. The lack of clarity and specificity in the evaluation methodology of the main text, makes it difficult to assess the soundness of the framework and its applicability in real-world scenarios. Based on these observations, I believe that further work is needed to improve the overall quality and relevance of the paper. The author should focus on refining their arguments, developing a more comprehensive evaluation methodology and providing more narrow messages.

**Update after comments from authors**

I changed my review from 5 to 7, following author's comments and revised version. See my comments below.

**Correctness:**

I have not noticed anything incorrect in the main text, but I haven't been into the (necessary) details of the supplementary material, and therefore cannot ensure that everything is scientifically sound.

**Documentation:**

It is not clear whether the contribution is meant to be a benchmark (i.e., software) or a dataset (i.e., text). From my understanding of the organisation of the repository, it seems to be the former, but there are several mentions of a dataset that are confusing (for instance, the presence of a data sheet in the supplementary material).

Otherwise, the repository is well-organised and allows for reproducibility, even if the source code lacks proper documentation and comments that could make the re-use by other reseach groups relatively difficult.

The use of GPT 4 and 3.5 raises two significant concerns regarding reproducibility that should be addressed. Firstly, it is not clear whether the tested versions (which are dated "March" according to the Supplementary Material) will be accessible to researchers from external sources, as only latest versions are generally accessible. This issue raises doubts about whether or not these experiments can be successfully replicated by other researchers. Secondly, to the best of my knowlege, access to GPT models is not free. While this does not invalidate the approach, it is important that the cost of reproducing the results is mentioned in the paper to ensure transparency and make sure that the findings and reproducibility are accessible to the broader scientific community.

**Ethics:**

No specific ethical concerns found, beyond the potential use by malicious actors as discussed in section "Limitations".

**Limitations:**

There is an important concern regarding the potential misuse of this work by malicious actors, who may use it to test the effectiveness of various strategies to bypass the safety mechanisms of language models. This possibility poses a risk for users of GPT-based systems. Despite this, my personal opinion is that the positive outcomes of publishing such frameworks outweigh the potential risks, as it facilitates scientific progress and allows for a deeper understanding of the field.

However, given the possible negative consequences associated with such work, I believe it would be valuable for the paper to include a discussion of this aspect. Such a discussion could help readers understand the reasoning behind the decision to publish the work and encourage greater discussion and consideration of responsible AI research.

**Opportunities For Improvement:**

The authors raised valid limitations concerning their work (Obscure pretraining data, subjectivity, specific focus on GPT models) that would deserve to appear in the main text with potential solutions to address them.

Overall, my main concern is about the significant amount of detail in the the supplementary material. While I appreciate the thoroghness of the analysis, it may be challenging for readers to fully grasp the significance and the soundness of the contribution without a significant investment of time and effort. As a reviewer, I understand that authors may feel limited by page restrictions. Nonetheless, in this case, some key information that should be in the main text is relegated to the supplementary material. This approach raises two issues for me:
1) insufficient detail in the main text to evaluate the scientific soundness;
2) a weak narrative of the paper, as each subsection is only connected by the trustworthiness concept, which is a bit shallow.

In other words, the paper seems like a collection of tests for evaluating different aspects of trustworthiness, where each aspect is taken independently to the others. This is also visible in the Related works section, where each aspect is considered separately. If we fully extend this reasoning, each subsection (or maybe a group of subsections) could be a separate contribution.

This does not put into question the quality of the contribution, and potential approaches to address the concerns mentioned earlier include:
1. Adopting a unified view to test different aspects jointly.
2. Studying the interplay between the different perspectives and how they interact with each other.
3. Discussing mitigation measures and exploring how they impact overall trustworthiness.
4. Connecting this framework with legal requirements in current AI regulations.
5. Expanding the benchmarking approach of the paper by testing other models, including open-source ones.

**Relation To Prior Work:**

This section is under-developed in the main text, and its inclusion in the Supplementary Material (where it is much more detailed) is not adequate.

**Summary And Contributions:**

The paper provides a comprehensive evaluation of the trustworthiness of recent large language models (LLMs), specifically GPT-3.5 and GPT-4. The authors examine various dimensions, including toxicity, stereotype bias, robustness, privacy, machine ethics, and fairness, and provide empirical findings on each. The paper highlights the new capabilities of LLMs to follow instructions and the potential concerns that arise from these capabilities. The findings are that GPT-4 is generally more trustworthy than GPT-3.5 but the authors also note instances where GPT-4 demonstrates higher toxicity than GPT-3.5. Overall, the paper aims to advance the field of LLMs by promoting the development of more reliable, unbiased, and transparent language models that meet the needs of users while upholding trustworthiness standards.

---

> ### Author Response · Authors · 2023-08-22
> **Thank you for your valuable comments**
>
> We thank the reviewer for the insightful comments and feedback! We are glad that the reviewer finds our work providing a comprehensive assessment of GPT models, analyzing a wide range of critical aspects, and making valuable findings to the scientific community and beyond. We provide detailed responses below.
>
> 1. “Adopting a unified view to test different aspects jointly.” “Studying the interplay between the different perspectives and how they interact with each other. / Expanding the benchmarking approach of the paper by testing other models, including open-source ones.”
>
> - Thank you for the valuable and constructive comments.  Following the reviewer’s suggestions, we have provided a unified view to test different aspects jointly by providing a unified testing API which allows the users to easily add new evaluation perspectives and evaluate different LLMs.
>
> - In particular, our unified testing API is able to evaluate different trustworthiness perspectives jointly through a single entry point. We have updated our evaluation framework in [Github](https://github.com/AI-secure/DecodingTrust). Our unified API features structured configuration, enabling the following functionality: (1), our API can provide users with a simple entry (main.py) to execute all experiments in one pass. (2), if you only want to run the selected evaluations of DecodingTrust (e.g., some trustworthiness perspectives), you can specify the argument to run specific scenarios (`python main.py +toxicity=toxic-gpt4`) in the command line input. (3), if you want to run the evaluations with your custom configuration, you can simply set up the sub-configuration file and override the corresponding argument in the command line input.
>
> - In addition, our API can evaluate different LLMs, including open LLMs hosted in Huggingface and proprietary LLMs through API queries.
> We also follow your suggestion and run the following top open-source LLMs in the Open LLM leaderboard, including Llama-v2-7B-Chat, Vicuna-7B, Alpaca-7B, MPT-7B, Falcon-7B, RedPajama, on all the trustworthiness perspectives. The results are shown below. For each perspective, we report one single score to facilitate clearer comparison, which is aggregated from the evaluation results of all scenarios under that perspective.  We have also added corresponding setup details, experimental results, and related analysis in Appendix L.
>
> *Table 1 Comprehensive evaluation results of open-source LLMs*
>
> | Model     | Toxicity | Stereotype Bias | Adversarial Robustness | OOD Robustness | Robustness to  Adv. Demonstrations | Privacy | Machine Ethics | Fairness |
> | --------- | -------- | --------------- | ---------------------- | -------------- | ---------------------------------- | ------- | -------------- | -------- |
> | Llama2    | 80.00    | 97.60           | 51.01                  | 75.65          | 55.54                              | 97.39   | 40.58          | 100.00   |
> | Vicuna    | 28.00    | 81.00           | 52.16                  | 59.10          | 57.99                              | 72.96   | 48.22          | 85.53    |
> | Alpaca    | 22.00    | 43.00           | 46.43                  | 51.79          | 34.15                              | 46.39   | 30.43          | 92.63    |
> | MPT       | 40.00    | 84.60           | 46.20                  | 64.26          | 58.25                              | 78.93   | 26.11          | 100.00   |
> | Falcon    | 39.00    | 87.00           | 43.98                  | 51.45          | 33.95                              | 70.26   | 50.28          | 100.00   |
> | RedPajama | 18.00    | 73.00           | 44.81                  | 54.21          | 58.51                              | 76.64   | 27.49          | 100.00   |
> | GPT-3.5   | 47.00    | 87.00           | 56.69                  | 73.58          | 81.28                              | 70.13   | 86.38          | 77.57    |
> | GPT-4     | 41.00    | 77.00           | 64.04                  | 87.55         | 77.94                              | 66.11   | 76.60          | 63.67    |
>
> We aim to study the interplay between the different perspectives based on the evaluations above. As illustrated in the table, among the 8 trustworthiness perspectives, GPT-4 achieves the best performance on 3 perspectives: Adversarial Robustness, Out-of-Distribution Robustness, and Robustness to Adversarial Demonstrations. The open-source model, Llama 2, achieves the best performance on 4 perspectives: Toxicity, Stereotype Bias, Privacy, and Fairness, which demonstrate the efforts that the Llama2 team have put into developing less-biased, privacy-aware, and fairness-aware LLMs. Overall, we can see that currently no model can achieve the best performance on all the perspectives, and there are tradeoffs between different perspectives. We believe our observations will lead to interesting future work on developing more trustworthy LLMs and understanding the underlying connections between different trustworthiness perspectives.

---

> ### Author Response · Authors · 2023-08-22
> **Thank you for your valuable comments**
>
> 2. “Discussing mitigation measures and exploring how they impact overall trustworthiness.”
>
> - Thanks for the insightful suggestion. We have listed several potential solutions and strategies to mitigate the potential vulnerabilities in the LLMs in Appendix M. Specifically, we mention four potential ways to safeguard the trustworthiness of LLMs, including:
>
>     -  **Safeguarding LLMs with additional knowledge and reasoning analysis**: As purely data-driven models such as GPT models would suffer from the imperfection of the training data and lack of reasoning capabilities in various tasks. Thus, it would be important to equip domain knowledge and logical reasoning capabilities for language models and safeguard their outputs to make sure they satisfy basic, domain knowledge and logic to ensure the trustworthiness of the model outputs.
>
>     - **Safeguarding LLMs based on consistency checking**: Our designed system prompts based on “role-playing" shows that models can be easily fooled based on role-changing and manipulation. This indicates that during the conversation of LLMs, it is possible to design diverse roles to ensure the consistency of the model’s answers, and therefore at least avoid the models being self-conflict. It is also possible to design different roles for the models to make sure it understands the context well to provide more informative and trustworthy answers.
>
>    - **Safeguarding LLMs via trustworthy finetuning**: Our generated challenging and adversarial prompts usually represent the long-taild and “rare” events of the original training data distribution. As a result, it would be helpful to use our generated challenging prompts to finetune the LLMs and improve their trustworthiness. On the other hand, we note that new adaptive adversarial attacks could still be conducted against the new finetuned LLMs, and therefore we need to be aware of new adaptive attacks and try to provide certain trustworthines verifications which are agnostic to actual attack strategies.
>
>     - **Verification for the trustworthiness of LLMs**: Empirical evaluations of LLMs are important but lack of guarantees, especially in safety-critical domains, so rigorous trustworthiness guarantees would be critical. To safeguard the trustworthiness of LLMs, it is important to provide verification for the trustworthiness of LLMs based on specific functionalities or properties, or map the discrete input space to their corresponding continuous space such as the embedding space with semantic preservation to perform verification leveraging existing verification tools in the continuous space.
>
> - We have incorporated the above discussions in our main text in the revision, and thank you for the valuable suggestions!
>
> 3. “Connecting this framework with legal requirements in current AI regulations.”
>
> - Thanks for the insightful suggestions. We follow your suggestions to extract the legal requirements from EU AI Act and the U.S. White House announcement. In summary, the trustworthiness of LLMs and other AI systems has become one of the key focuses of policymakers, such as the European Union's Artificial Intelligence Act (AIA) and the United States AI Bill of Rights. The AIA adopts a risk-based approach that categorizes AI systems based on their risk levels, necessitating stringent compliance assessments for high-risk systems. The U.S. has proposed principles for safe AI systems, including safety, fairness, privacy, and human-in-the-loop intervention.
> These regulations align well with the trustworthiness perspectives that we define and evaluate in our DecodingTrust platform, such as adversarial robustness, out-of-distribution robustness, privacy, fairness, and disclosure. In addition, these regulatory measures reflect growing attention on evaluation and risk assessment for AI systems from different angles, which resonates with the research community and our research efforts to ensure responsible and reliable AI evaluation and deployment. We believe our platform will help facilitate the standard evaluation and risk assessment benchmarking efforts for AI systems and contribute to developing trustworthy ML and AI systems in practice.
>
> - Moreover, as shown in our evaluation of GPT models and open LLMs, none of them can achieve the best performance on all the trustworthiness perspectives, which provides an insightful understanding of different models and the underlying connections between different trustworthiness perspectives. We believe our evaluation can shed light on the development of trustworthy LLMs considering different perspectives. We have incorporated the detailed discussion highlighted in blue in Appendix Q in the revision.

---

> ### Author Response · Authors · 2023-08-22
> **Thank you for your valuable comments**
>
> 4. “Given the possible negative consequences associated with such work, I believe it would be valuable for the paper to include a discussion of this aspect. Such a discussion could help readers understand the reasoning behind the decision to publish the work and encourage greater discussion and consideration of responsible AI research.”
>
> - Thanks for the insightful comments. We follow your suggestions and add a more detailed discussion about potential negative consequences such as the misuse of the information in Appendix N to make sure users are aware of the potential negative impacts. We agree that such discussion will help readers to understand the reasoning behind the decision to publish the work and we will put more detailed discussion below for such reasoning and our motivation of the work.
>
> - First, we believe sharing the high-level settings of our evaluation will be beneficial for both researchers and practitioners who aim to train LLMs and understand the model capabilities, and need to be aware of the model vulnerabilities before deployment. In particular, our platform and findings provide comprehensive evaluations to understand the model capabilities and vulnerabilities, which is critical before deploying LLMs in practice. Similar to several concurrent efforts in exploring the vulnerabilities of LLMs [1,2,3], we aim to provide better understandings and insights about the models in adversarial environments, so that users could avoid the potential attacks.
>
> - In addition, for the detailed generated challenging and adversarial prompts, we will share the prompts used in our evaluation, while preserving some newly generated prompts for future evaluation to balance between the research openness and information misuse. In particular, we will keep parts of the newly generated challenging data private to ensure that as a white-hat model evaluation, we are slightly ahead of the actual adversaries in the real world, so that we can start to design potential solutions against these vulnerabilities before they are implemented in practice.
>
> [1] Qiu, H., Zhang, S., Li, A., He, H., & Lan, Z. (2023). Latent Jailbreak: A Benchmark for Evaluating Text Safety and Output Robustness of Large Language Models. ArXiv, abs/2307.08487.
>
> [2] Liu, Y., Yao, Y., Ton, J., Zhang, X., Cheng, R.G., Klochkov, Y., Taufiq, M.F., & Li, H. (2023). Trustworthy LLMs: a Survey and Guideline for Evaluating Large Language Models' Alignment.
>
> [3] https://www.jailbreakchat.com/
>
> 5. “Relation To Prior Work: This section is under-developed in the main text, and its inclusion in the Supplementary Material (where it is much more detailed) is not adequate.”
>
> - Thanks for the valuable comments. We follow your suggestions and add a more detailed related work section in Section 11 of the main paper to provide more background about the trustworthiness of LLMs, including existing benchmarks, adversarial strategies against LLMs, and related regulations. We have highlighted our revision in blue. Thank you for the valuable suggestions.
>
> 6. “It is not clear whether the contribution is meant to be a benchmark (i.e., software) or a dataset (i.e., text). From my understanding of the organization of the repository, it seems to be the former, but there are several mentions of a dataset that are confusing (for instance, the presence of a data sheet in the supplementary material). Otherwise, the repository is well-organized and allows for reproducibility, even if the source code lacks proper documentation and comments that could make the re-use by other research groups relatively difficult.”
>
> - Thanks for the valuable comments. Indeed, our DecodingTrust project provides both a unified trustworthiness evaluation platform and several challenging and adversarial datasets for trustworthiness evaluation on different LLMs flexibly.
>
> - To provide a unified trustworthiness evaluation platform, we allow users to use a simple entry (`python main.py`) to execute all the evaluations for different perspectives and different models in one pass. We also provide an [updated tutorial documentation] (https://github.com/AI-secure/DecodingTrust/blob/main/Tutorial.md) to help users to walk through our codebase and allow flexible customization to configure the model parameters and evaluation settings.
>
> - To provide challenging evaluation datasets, we propose and generate new challenging user prompt datasets that can better, for instance, elicit model toxicity than existing standard benchmarks. We have added the related discussions in Section 1 in the revision to make it more clear.

---

> ### Author Response · Authors · 2023-08-22
> **Thank you for your valuable comments**
>
> 7. “The use of GPT 4 and 3.5 raises two significant concerns regarding reproducibility that should be addressed. Firstly, it is not clear whether the tested versions (which are dated "March" according to the Supplementary Material) will be accessible to researchers from external sources, as only latest versions are generally accessible. This issue raises doubts about whether or not these experiments can be successfully replicated by other researchers. Secondly, to the best of my knowledge, access to GPT models is not free. While this does not invalidate the approach, it is important that the cost of reproducing the results is mentioned in the paper to ensure transparency and make sure that the findings and reproducibility are accessible to the broader scientific community.”
>
> - Thanks for the insightful comments. According to the [official documentation from OpenAI](https://platform.openai.com/docs/deprecations/), GPT-3.5 and GPT-4 are released with incremental updates, without immediate deprecation of older versions. The March-released GPT-3.5 (`gpt-3.5-turbo-0301`) and GPT-4 (`gpt-4-0314`) models we used in our evaluation are still available for external researchers until June 13, 2024, which ensures the reproducibility. Moreover, we have open-sourced all of our code and corresponding model outputs in the GitHub repo to make sure that our results are reproducible.
> - Furthermore, our codebase can support the evaluation of open-source LLMs (e.g., models hosted in Huggingface), which ensures that our evaluation framework and findings are reproducible and accessible to the broader scientific community. (Table 1 above shows our evaluations on different open-source LLMs and GPT models.)
> - Regarding the cost of the evaluation, we have added details of the computation and query costs in Appendix K in the revision following the suggestions. We also put a detailed analysis of costs in Table 2 below.
>
> Table 2 summarizes the 1) total number of prompts, 2) total number of prompt tokens, 3) total number of completion tokens, and 4) total run costs for each trustworthiness perspective and GPT models.
>
> *Table 2. Total computing costs of evaluation on different trustworthiness perspectives on GPT models*
> | Perspectives                                 | Models  | #/ Prompts | #/ Prompt Tokens | #/ Completion Tokens | Total Cost ($) |
> | -------------------------------------------- | ------- | ---------- | ---------------- | -------------------- | -------------- |
> | Toxicity                                     | GPT-3.5 | 49,200      | 10,966,554       | 15,796,800           | 78.14          |
> | Toxicity                                     | GPT-4   | 49,200      | 10,966,554       | 15,796,800           | 2158.97        |
> | Stereotype                                   | GPT-3.5 | 3,456       | 766,296          | 12,960,000           | 27.46          |
> | Stereotype                                   | GPT-4   | 3,456       | 766,296          | 12,960,000           | 800.58         |
> | Adversarial Robustness                       | GPT-3.5 | 42,755      | 3,596,216        | 684,080              | 9.30            |
> | Adversarial Robustness                       | GPT-4   | 42,755      | 3,596,216        | 684,080              | 162.23         |
> | OOD                                          | GPT-3.5 | 47,079      | 13,879,675       | 470,790              | 28.70           |
> | OOD                                          | GPT-4   | 47,079      | 13,879,675       | 470,790              | 444.64         |
> | Robustness against adversarial demonstration | GPT-3.5 | 233,100     | 152,882,443      | 322,259              | 306.41         |
> | Robustness against adversarial demonstration | GPT-4   | 233,100     | 144,558,043      | 256,140              | 4352.11        |
> | Privacy                                      | GPT-3.5 | 106,150    | 6,363,542        | 2,408,800            | 17.54          |
> | Privacy                                      | GPT-4   | 106,150    | 6,363,542        | 2,408,800            | 335.43         |
> | Machine Ethics                               | GPT-3.5 | 21,869     | 6,796,656        | 373,380              | 15.31          |
> | Machine Ethics                               | GPT-4   | 21,869     | 6,796,656        | 373,380              | 242.29         |
> | Fairness                                     | GPT-3.5 | 32,400     | 16,798,525       | 180,000              | 34.00             |
> | Fairness                                     | GPT-4   | 32,400     | 16,798,525       | 180,000              | 503.35         |

---

> ### Author Response · Authors · 2023-08-22
> **Thank you for your valuable comments**
>
> - In addition, we further break down our evaluation for each perspective into detailed scenarios, and Table 46-53 in Appendix K show the similar computing and query costs for these detailed scenarios under different trustworthiness perspectives. Similarly, we report the 1) number of prompts for each scenario and GPT models, 2) number of tokens of the prompts, 3) number of completion tokens that answer the prompts, 4) single run cost of answering the prompts, 5) number of the run repetitions, and 6) total run cost. These tables also allow users to flexibly determine whether they want to run some subsets of the scenarios of each perspective based on their available resources.
>
> - We thank the reviewers for all the insightful suggestions and comments, and hope our additional experiments and discussions will help address your concerns. We are very grateful for the reviewer’s suggestions on helping to improve our work.

---

> > ### Comment · Reviewer_qCen · 2023-08-25
> > **Comments on the response to the review**
> >
> > Thank you very much for taking the time to respond to my comments, and to provide a revised version of the manuscript. I appreciate the efforts made to improve the contribution and address my concerns.
> >
> > I still do believe that the manuscript is too substantial (24 pages of main text + 100 pages of supplementary material) to be correctly evaluated, limiting the discussion and engagement with authors on each specific points discussed in the paper. Nonetheless, I think that the the overall approach and its implementation is serious and consistent enough to be relevant to many researchers and practitionners in the field. In particular, it seems relatively feasible to reproduce the results on specific aspects, and build on top of it more focused analysis of the evaluation strategies and improve them.
> >
> > Based on that, I will update my initial review to take into account the revision of the manuscript.

---

> > > ### Author Response · Authors · 2023-08-29
> > > **Thank you for your valuable follow-up comments**
> > >
> > > Thank you so much for your valuable suggestions. We provide rich content in the appendix due to the comprehensiveness of our trustworthiness evaluations as recognized by the reviewers. Following your suggestions, we have made sure that the main paper (10 pages) is self-contained, and readers can clearly get the evaluation goal, setup, and findings for each perspective by only reading the main paper. In the appendix, we mainly add more examples, conversational templates, and some experimental details. We would like to note that these detailed examples will help readers to get more intuition and understanding about the vulnerabilities and capabilities of LLMs, but they are definitely not required in order to understand the paper. For instance, our related work in the appendix is of 6 pages to provide a very comprehensive literature survey, which we believe will largely benefit the community, but it is not required to read the entire related work to understand our paper.
> > >
> > > In addition, following the reviewer’s suggestion, we have further moved some examples and conversational templates from the appendix. We hope this addresses your concerns. Please let us know if you have other suggestions or comments, and we are glad to discuss more and further improve our work to contribute to the community! Thank you for your constructive feedback again!!

---

### Official Review · Reviewer_j1j9 · 2023-07-21
**Detailed analysis of current challenges in LLMs, with a thorough problem description.**

**Rating:** 10
**Confidence:** 4
**Correctness:** The paper appears to be correct.
**Clarity:** The paper is well-written and easy to…

**Strengths:**

In my opinion, the paper successfully identifies most of the important known threats and conducts thorough experiments using useful datasets. The threats are described comprehensively and highlight the current challenges of LLMs and you can clearly see that a lot of effort has been put into the paper and the benchmark itself.

**Additional Feedback:**

No additional feedback

**Documentation:**

The project is well documented and include a website, and detailed instructions and code for every aspect.

**Ethics:**

There are ethical aspects in the paper, but they are clearly indicated

**Limitations:**

The benchmark is not easy to extend without a lot of manual effort. Each method and dataset are manually selected or crafted. However, I can clearly see the difficulties here and appreciate the effort by the authors.

The evaluation is limited to two (recent) LLMs

**Opportunities For Improvement:**

The paper does not specifically describe the benchmark framework. What makes it a benchmark? How easy is it to add new evaluation aspects?

What are other attack vectors in case of, for example, whitebox access to a model? This is particularly important to evaluate a worst-case scenario.

**Relation To Prior Work:**

The paper discusses significant related work.

**Summary And Contributions:**

The paper provides a comprehensive evaluation of the security, privacy, and trustworthiness of language models, with a focus on GPT-3.5 and GPT-4. It covers topics such as toxicity, stereotypes, adversarial robustness, and information leakage, among others. For each aspect, the authors provide a detailed explanation and introduction to the dataset used, tailored specifically to the analyzed task. The results are presented for various facets of each aspect and include multiple datasets.

---

> ### Author Response · Authors · 2023-08-22
> **Thank you for your valuable comments**
>
> We thank the reviewer for the valuable suggestions and comments! We are glad that the reviewer finds our paper providing comprehensive evaluation, successfully identifying most of the important known threats, and conducting thorough experiments with insightful findings. We will provide detailed responses below.
>
> 1. “The paper does not specifically describe the benchmark framework. What makes it a benchmark? How easy is it to add new evaluation aspects”
>
> - Thanks for the valuable questions and feedbacks. We indeed aim to make the DecodingTrust a general benchmark framework as suggested, which is easy to add new evaluation perspectives and easy to evaluate other models. In particular, we follow the principles of constructing standard benchmarks [1, 2] to ensure that our platform can serve as a benchmark for evaluating the trustworthiness of language models. Below we highlight some key principles of and illustrate how DecodingTrust aligns with these principles.
>   - *(1) Standardization*: Our benchmark designs leverages a set of standardized criteria and metrics that are used consistently across prior studies on LLM evaluation and different models.
>   - *(2) Accessibility and Coverage*: DecodingTrust is designed to facilitate comparisons among different LLMs conveniently. It is easy to evaluate any LLM by providing the model URL to the DecodingTrust platform following our tutorial. In particular, for each trustworthiness perspective, we have developed multiple standardized evaluation structures (each includes a detailed setup, datasets, metrics, etc), as shown in Figure 3 in Appendix A. This set of standardized scenarios serves the purpose of evaluating the trustworthiness of different language models. In the revision, we also integrate six more open-source LLMs from the top Open LLM leaderboard to analyze the trustworthiness of open-source LLMs. We include our evaluation results in Appendix L in the revision, which shed light on the future directions of improving open-source LLMs. In summary, these standardized scenarios can be effectively utilized to compare the performance and resilience of different language models across various trustworthiness perspectives.
>    - *(3) Disincentives for Biased Models*: DecodingTrust benchmark conducts a comprehensive and reliable measure of different trustworthiness perspectives for LLMs, which further provides insightful understandings of the capabilities and vulnerabilities of LLMs. In particular, practitioners have proposed applying capable GPT models to sensitive applications such as healthcare and finance, where mistakes can be costly. Hence, trustworthiness is a crucial aspect that requires attention. DecodingTrust manages to provide a comprehensive trustworthiness-focused evaluation on GPT models together with some open-source models, assessing their performance and resilience in adversarial environments across different aspects of trustworthiness. Thus, we believe DecodingTrust will serve as a standard benchmark to disincentive untrustworthy models especially for safety-critical domains in practice.
>    - *(4) Documentation and Transparency*: A benchmark should have clear documentation that outlines the methodology, procedures, and metrics used in the evaluation process. DecodingTrust is well documented and includes a website, detailed instructions and codes for every trustworthiness perspective.
>
>
> - In addition, to add new evaluation perspectives, the overhead is very low thanks to our unified DecodingTrust API. In particular, users only need to specify the model name (either an OpenAI model or a Huggingface Hub model) and the perspective they would like to evaluate in the command line interface following our tutorial. The code base of DecodingTrust consists of a wrapper that unifies the programming interface of OpenAI API and Huggingface Hub models, the Python modules that host the implementation of each trustworthiness perspective, a configuration system backed by Hydra [3], and a centralized code runner that delegates command line input to perform the evaluation.
> By adding the Python module of the new perspective following the unified format, we could easily conduct evaluations on the new perspective.
>
> [1] [What Will it Take to Fix Benchmarking in Natural Language Understanding?](https://arxiv.org/pdf/2104.02145.pdf)
>
> [2] [Datasheets for Datasets](https://arxiv.org/pdf/1803.09010.pdf)
>
> [3] [Hydra - A framework for elegantly configuring complex applications](https://github.com/facebookresearch/hydra)

---

> ### Author Response · Authors · 2023-08-22
> **Thank you for your valuable comments**
>
> 2. “What are other attack vectors in case of, for example, whitebox access to a model? This is particularly important to evaluate a worst-case scenario.”
>
> - Thank you for the insightful question. We agree that conducting whitebox attacks against LLMs can serve as a worst-case evaluation. On the one hand, since both GPT-3.5 and GPT-4 only provide text completion without providing direct access to model weights or output logits, it can be challenging to directly conduct whitebox attacks on GPT-3.5 and GPT-4. On the other hand, **we have conducted whitebox attacks on open-source LLMs (Alpaca, Vicuna, and Stable Vicuna)** to evaluate their worst-case performance. Moreover, we transfer our adversarial textural attack instances generated by whitebox attacks to GPT-3.5 and GPT-4 as AdvGLUE++, and we find that **our whitebox-generated adversarial textural attacks against open-source LLMs can effectively transfer and attack GPT-3.5 and GPT-4 models**. For example, the robust accuracy of GPT-3.5 and GPT-4 significantly drop on our adversarial instances generated against Alpaca-7B, where the corresponding robust accuracies of GPT-3.5 and GPT-4 are only 49.23% and 55.64%, respectively.

---

> ### Author Response · Authors · 2023-08-22
> **Thank you for your valuable comments**
>
> 3. "The benchmark is not easy to extend without a lot of manual effort. Each method and dataset are manually selected or crafted. However, I can clearly see the difficulties here and appreciate the effort by the authors. The evaluation is limited to two (recent) LLMs."
>
> - We deeply appreciate your valuable comments. We are pleased to note that our project code has been unified as a unified API, which is easy to extend (e.g., adding new evaluation perspectives), and can be used to evaluate any open-source models conveniently.
>
> - In particular, our unified DecodingTrust API has the following characteristics:
>      - **1) Structured and unified.** As for the API configuration, the top-level main configuration file contains basic information required by all perspectives such as model name and API key, and the sub-configuration files contain the dedicated configurations for each trustworthy perspective, respectively. Based on this structured configuration, **first of all**, our API can provide users with a simple entry (`python main.py`) to execute all experiments in one pass. **Second of all**, if the users only want to run the selected evaluations of DecodingTrust (e.g., some trustworthiness perspectives), they can specify the argument to run specific scenarios (`python main.py +toxicity=realtoxicityprompts-toxic`) in the command line input. **Lastly**, if the users want to run the evaluations with their custom configuration, they can simply set up the sub-configuration file and override the corresponding argument in the command line input. We have also updated our README as well as a [tutorial page](https://github.com/AI-secure/DecodingTrust/blob/main/Tutorial.md) for more examples.
>     - **2) Incorporating more open-source language models.**
> Our API can support interacting with different API-based LLMs as well as Huggingface open LLMs. Thus, practitioners can utilize our DecodingTrust API to evaluate their own LLMs by simply configuring the corresponding argument in the command line input following our tutorial.
>
> - We have also followed your suggestion and evaluate the following top open-source LLMs in the Open LLM leaderboard, including Llama-v2-7B-Chat, Vicuna-7B, Alpaca-7B, MPT-7B, Falcon-7B,  RedPajama, on all the trustworthiness perspectives. The results are shown below. For each perspective, we report one single score to facilitate clearer comparison, which is aggregated from the evaluation results of all scenarios under that perspective.  We have also added corresponding setup details, aggregation protocols, experimental results, and related analysis in Appendix L.
>
> *Table 1 Comprehensive evaluation results of open-source LLMs*
>
> | Model     | Toxicity | Stereotype Bias | Adversarial Robustness | OOD Robustness | Robustness to  Adv. Demonstrations | Privacy | Machine Ethics | Fairness |
> | --------- | -------- | --------------- | ---------------------- | -------------- | ---------------------------------- | ------- | -------------- | -------- |
> | Llama2    | 80.00    | 97.60           | 51.01                  | 75.65          | 55.54                              | 97.39   | 40.58          | 100.00   |
> | Vicuna    | 28.00    | 81.00           | 52.16                  | 59.10          | 57.99                              | 72.96   | 48.22          | 85.53    |
> | Alpaca    | 22.00    | 43.00           | 46.43                  | 51.79          | 34.15                              | 46.39   | 30.43          | 92.63    |
> | MPT       | 40.00    | 84.60           | 46.20                  | 64.26          | 58.25                              | 78.93   | 26.11          | 100.00   |
> | Falcon    | 39.00    | 87.00           | 43.98                  | 51.45          | 33.95                              | 70.26   | 50.28          | 100.00   |
> | RedPajama | 18.00    | 73.00           | 44.81                  | 54.21          | 58.51                              | 76.64   | 27.49          | 100.00   |
> | GPT-3.5   | 47.00    | 87.00           | 56.69                  | 73.58          | 81.28                              | 70.13   | 86.38          | 77.57    |
> | GPT-4     | 41.00    | 77.00           | 64.04                  | 87.55         | 77.94                              | 66.11   | 76.60          | 63.67    |
>
> From the table, GPT-4 achieves the best performance on 3 perspectives: Adversarial Robustness, Out-of-Distribution Robustness, and Robustness to Adversarial Demonstrations. The open-source model, Llama 2, achieves the best performance on 4 perspectives: Toxicity, Stereotype Bias, Privacy, and Fairness, which demonstrate the efforts that the Llama2 team has put into developing less-biased, privacy-aware, and fairness-aware LLMs. On the other hand,  we can see that currently, no model can achieve satisfactory performance for all the perspectives. In light of these observations, developing more trustworthy LLMs remains an important task for future work. We appreciate your comments to help improve our work.

---

> > ### Comment · Reviewer_j1j9 · 2023-08-23
> > **Response to Comment**
> >
> > Thank you for the detailed clarifications and I also appreciate the effort in changing the paper according to it. I am still thinking that your work is a great contribution to the community and to study the trustworthiness of LLMs. I will keep my score and am certain that this will positively affect the overall result for you : )

---

### Official Review · Reviewer_UhDT · 2023-07-27
**Comprehensive evaluations of GPT 3.5 and GPT 4**

**Rating:** 7
**Confidence:** 4
**Correctness:** The paper is sound
**Clarity:** The paper is very clear

**Strengths:**

Strengths:
- The problem of systematic evaluation of trustworthiness is very important and somewhat urgent, this paper provides a valuable and timely contribution
- The evaluation of every trustworthiness aspect of GPT models is very thorough and with detailed takeaways providing insight into the trustworthiness of GPT models and evaluating their robustness under potential misuse. The insights from this paper highlight opportunities for LLM improvement and areas where more research is needed to improve LLM trustworthiness.
- the systematic trustworthiness evaluation benchmark is a valuable contribution in itself, in addition to the insights about GPT models
- the dataset and evaluation scripts are released and available to the public under permissive license
- the paper is very clear and enjoyable to read


**Additional Feedback:**

Questions:
- The total compute is not specified. Can you provide any metrics on how long it took to run the evaluation and how much compute resources were used (e.g. maybe in the form of the number of processed tokens by the OpenAI API)?

**Documentation:**

Yes, details on the dataset and evaluation scripts are provided in the supplement, on the website and on GitHub

**Ethics:**

I do not see any ethical concerns

**Limitations:**

Limitations are addressed in appendix

**Opportunities For Improvement:**

Weaknesses:
- The paper only focuses on GPT models, however, those are proprietary models. It would be valuable to add evaluation of open-source models available to the general public and identify the most trustworthy open models.
- While Figure 1 is excellent and the paper is easy to read, it lacks sufficient explanations of experimental setups for most sections of the main body and refers to appendix for crucial details and explanations central to the experimental setup. It would be great to provide necessary details in the main body and refer to appendix only for supplementary details.


**Relation To Prior Work:**

Prior work is covered well

**Summary And Contributions:**

The paper provides comprehensive evaluation of trustworthiness of GPT models across diverse aspects ranging from toxicity to adversarial robustness, to machine ethics and fairness, to privacy, and others. In total, the paper explores 8 evaluation dimensions. The evaluation is experimental and very thorough, with insightful takeaways. The dataset and evaluation scripts are released under permissive licenses and available.

---

> ### Author Response · Authors · 2023-08-22
> **Thank you for your valuable comments**
>
> 1. “The paper only focuses on GPT models, however, those are proprietary models. It would be valuable to add evaluation of open-source models available to the general public and identify the most trustworthy open models.”
>
> - Thanks for your valuable suggestions on expanding the scope of our work! We are pleased to note that our project code has been unified as a unified API, which can be used to evaluate any open-source models.
>
> - In particular, our unified DecodingTrust API has the following characteristics:
>      - **1) Structured and unified.** As for the API configuration, the top-level main configuration file contains basic information required by all perspectives such as model name and API key, and the sub-configuration files contain the dedicated configurations for each trustworthy perspective, respectively. Based on this structured configuration, **first of all**, our API can provide users with a simple entry (`python main.py`) to execute all experiments in one pass. **Second of all**, if the users only want to run the selected evaluations of DecodingTrust (e.g., some trustworthiness perspectives), they can specify the argument to run specific scenarios (`python main.py +toxicity=realtoxicityprompts-toxic`) in the command line input. **Lastly**, if the users want to run the evaluations with their custom configuration, they can simply set up the sub-configuration file and override the corresponding argument in the command line input. Overall, the unified API offers users a convenient and user-friendly approach to effortlessly conduct evaluations according to their preferred settings. We have also updated our README as well as a [tutorial page](https://github.com/AI-secure/DecodingTrust/blob/main/Tutorial.md) for more examples.
>     - **2) Incorporating more open-source language models.**
> Our API can support interacting with different API-based LLMs as well as Huggingface open LLMs. Thus, practitioners can utilize our DecodingTrust API to evaluate their own LLMs by simply configuring the corresponding argument in the command line input following our tutorial. We have also followed your suggestion and evaluate the following top open-source LLMs in the Open LLM leaderboard, including Llama-v2-7B-Chat, Vicuna-7B, Alpaca-7B, MPT-7B, Falcon-7B,  RedPajama, on all the trustworthiness perspectives. The results are shown below. For each perspective, we report one single score to facilitate clearer comparison, which is aggregated from the evaluation results of all scenarios under that perspective.  We have also added corresponding setup details, aggregation protocols, experimental results, and related analysis in Appendix L.
>
> *Table 1 Comprehensive evaluation results of open-source LLMs*
>
> | Model     | Toxicity | Stereotype Bias | Adversarial Robustness | OOD Robustness | Robustness to  Adv. Demonstrations | Privacy | Machine Ethics | Fairness |
> | --------- | -------- | --------------- | ---------------------- | -------------- | ---------------------------------- | ------- | -------------- | -------- |
> | Llama2    | 80.00    | 97.60           | 51.01                  | 75.65          | 55.54                              | 97.39   | 40.58          | 100.00   |
> | Vicuna    | 28.00    | 81.00           | 52.16                  | 59.10          | 57.99                              | 72.96   | 48.22          | 85.53    |
> | Alpaca    | 22.00    | 43.00           | 46.43                  | 51.79          | 34.15                              | 46.39   | 30.43          | 92.63    |
> | MPT       | 40.00    | 84.60           | 46.20                  | 64.26          | 58.25                              | 78.93   | 26.11          | 100.00   |
> | Falcon    | 39.00    | 87.00           | 43.98                  | 51.45          | 33.95                              | 70.26   | 50.28          | 100.00   |
> | RedPajama | 18.00    | 73.00           | 44.81                  | 54.21          | 58.51                              | 76.64   | 27.49          | 100.00   |
> | GPT-3.5   | 47.00    | 87.00           | 56.69                  | 73.58          | 81.28                              | 70.13   | 86.38          | 77.57    |
> | GPT-4     | 41.00    | 77.00           | 64.04                  | 87.55         | 77.94                              | 66.11   | 76.60          | 63.67    |
>
> From the table, GPT-4 achieves the best performance on 3 perspectives: Adversarial Robustness, Out-of-Distribution Robustness, and Robustness to Adversarial Demonstrations. The open-source model, Llama 2, achieves the best performance on 4 perspectives: Toxicity, Stereotype Bias, Privacy, and Fairness, which demonstrate the efforts that the Llama2 team has put into developing less-biased, privacy-aware, and fairness-aware LLMs. On the other hand,  we can see that currently, no model can achieve satisfactory performance for all the perspectives. In light of these observations, developing more trustworthy LLMs remains an important task for future work.

---

> ### Author Response · Authors · 2023-08-22
> **Thank you for your valuable comments**
>
> 2. “While Figure 1 is excellent and the paper is easy to read, it lacks sufficient explanations of experimental setups for most sections of the main body and refers to appendix for crucial details and explanations central to the experimental setup. It would be great to provide necessary details in the main body and refer to appendix only for supplementary details.”
>
> - Thanks for your understanding that it is challenging for us to cover all detailed information in the current 9-page paper, and thanks for the reminder to keep the last version self-read and self-contained, which is really important and helpful for readers. To make the main paper self-contained following the suggestion, we leverage the additional one page in the final version to make sure that the following items are included in each trustworthiness perspective for consistency and completeness.
>     - Goal: This part allows readers easily understand what we aim to evaluate and what scenarios will be included.
>     - Setup: This part includes 1) the task we are focusing on, 2) the scenarios we will design and evaluate, 3) the descriptions of the datasets, and 4) the evaluation metrics.
>     - Results and findings: All the figures and tables in this part are also self-contained, whose title will clearly convey the content and purpose of the figure or table without requiring readers to refer to the main text.
> - For instance, we have added the details of the stereotype dataset (e.g., the selected stereotype groups and stereotype topics, and some examples of our user and system prompts) in the revision Section 3. We have also added more explanations about the aspects that OOD robustness aims to evaluate (i.e., OOD knowledge, OOD language style, and OOD in-context demonstrations) in revision Section 5.

---

> ### Author Response · Authors · 2023-08-22
> **Thank you for your valuable comments**
>
> 3. “The total compute is not specified. Can you provide any metrics on how long it took to run the evaluation and how much compute resources were used (e.g. maybe in the form of the number of processed tokens by the OpenAI API)”
>
> - Thanks for your valuable suggestions about demonstrating the computing resources used in the evaluations in the paper. We have added detailed computing and query costs and analysis in Appendix K in the revision following the suggestions.
>
> - Table 2 summarizes the 1) total number of prompts, 2) total number of prompt tokens, 3) total number of completion tokens, and 4) total run costs for each trustworthiness perspective and each GPT model.
>
> *Table 2. Total computing costs of evaluation on different trustworthiness perspectives on GPT models*
> | Perspectives                                 | Models  | #/ Prompts | #/ Prompt Tokens | #/ Completion Tokens | Total Cost ($) |
> | -------------------------------------------- | ------- | ---------- | ---------------- | -------------------- | -------------- |
> | Toxicity                                     | GPT-3.5 | 49,200      | 10,966,554       | 15,796,800           | 78.14          |
> | Toxicity                                     | GPT-4   | 49,200      | 10,966,554       | 15,796,800           | 2158.97        |
> | Stereotype                                   | GPT-3.5 | 3,456       | 766,296          | 12,960,000           | 27.46          |
> | Stereotype                                   | GPT-4   | 3,456       | 766,296          | 12,960,000           | 800.58         |
> | Adversarial Robustness                       | GPT-3.5 | 42,755      | 3,596,216        | 684,080              | 9.30            |
> | Adversarial Robustness                       | GPT-4   | 42,755      | 3,596,216        | 684,080              | 162.23         |
> | OOD                                          | GPT-3.5 | 47,079      | 13,879,675       | 470,790              | 28.70           |
> | OOD                                          | GPT-4   | 47,079      | 13,879,675       | 470,790              | 444.64         |
> | Robustness against adversarial demonstration | GPT-3.5 | 233,100     | 152,882,443      | 322,259              | 306.41         |
> | Robustness against adversarial demonstration | GPT-4   | 233,100     | 144,558,043      | 256,140              | 4352.11        |
> | Privacy                                      | GPT-3.5 | 106,150    | 6,363,542        | 2,408,800            | 17.54          |
> | Privacy                                      | GPT-4   | 106,150    | 6,363,542        | 2,408,800            | 335.43         |
> | Machine Ethics                               | GPT-3.5 | 21,869     | 6,796,656        | 373,380              | 15.31          |
> | Machine Ethics                               | GPT-4   | 21,869     | 6,796,656        | 373,380              | 242.29         |
> | Fairness                                     | GPT-3.5 | 32,400     | 16,798,525       | 180,000              | 34.00             |
> | Fairness                                     | GPT-4   | 32,400     | 16,798,525       | 180,000              | 503.35         |
>
> - In addition, we further break down our evaluation for each perspective into detailed scenarios, and Table 46-53 in Appendix K show the similar computing and query costs for these detailed scenarios under different trustworthiness perspectives. Similarly, we report the 1) number of prompts for each scenario and GPT models, 2) number of tokens of the prompts, 3) number of completion tokens that answer the prompts, 4) single run cost of answering the prompts, 5) number of the run repetitions, and 6) total run cost. These tables also allow users to flexibly determine whether they want to run some subsets of the scenarios of each perspective based on their available

---

> ### Author Response · Authors · 2023-08-22
> **Thank you for your valuable comments**
>
> We thank the reviewer once again for the valuable suggestions and comments! We are glad that the reviewer finds our paper providing a comprehensive, valuable, and timely evaluation, conducting very thorough experiments with detailed takeaways, and very clear and enjoyable to read. We have provided detailed responses above.

---

### Official Review · Reviewer_JNyJ · 2023-07-28
**An extensive study on Trustworthiness in GPT models**

**Rating:** 7
**Confidence:** 4
**Correctness:** the submission and claims seem correc…

**Strengths:**

1- different evaluation metrics and different capabilities of GPT models are evaluated relying on different standard benchmarks.
2- a benchmark for toxiticity and trustworthiness is presented
3- an extensive study for different aspects are presented with a profound analysis

**Additional Feedback:**

There is a concern or more as a question to the authors:
- Was it expensive to run all these prompts on GPT-4?  is a normal user or a student able to run all these prompts? If you can mention this in the paper, that would be really apprecoated.

**Clarity:**

It is well written, the only point for me is that the authors want to compact all the experiments and results in the 9 pages, which sometimes leads to going to the appendix a lot and make it hard to follow. Other than that, the paper is very well presented and it is well organized.

**Documentation:**

They are providing a url which ahs all their code snippets and the datasets they used.

**Ethics:**

no, I do not suspect that.

**Limitations:**

yes, the authors are adressing the potentail social impact by  their evaluation and study.

**Opportunities For Improvement:**

The details of the stereotype dataset should be in the paper not the appendix, this is important detail and if the appendix is separated later in the proceeding, it would be hard for the reader to follow.
Fig.2 is a bit misleading without its illustration which is also in the appendix. (hard to follow, either remove the parts, or add the illustration with it)
out-of-distribution robustness should be illustrated in what perspective are we evaluating, that was not clear to me.
In general, the appendix contained many important information, which made the written harder to follow without. I know you want to keep all the info inside the 9 pages, but consider in the last version to keep it self-read and self-contained.


**Relation To Prior Work:**

Yes in the introduction, they are relating to the prior contributions and they are relying on many prior benchmarks in their studies.

**Summary And Contributions:**

The paper presents a very intensive and extensive study on trustworthiness in GPT models, they are evaluating different perspectives of trustworthiness using different benchmarks. This is not mainly a benchmark paper but kind of papers we need to understand important perspectives in new models. This paper contributes to our understandings of how GPT models treat the toxic and trustworthiness prompts.

---

> ### Author Response · Authors · 2023-08-22
> **Thank you for your valuable comments**
>
> We thank the reviewer for the valuable suggestions and comments! We are glad that the reviewer finds our paper providing a comprehensive evaluation, and conducting extensive experiments with profound analysis. We will provide detailed responses below.
>
> 1. "The details of the stereotype dataset should be in the paper not the appendix, this is important detail and if the appendix is separated later in the proceeding, it would be hard for the reader to follow. Fig.2 is a bit misleading without its illustration which is also in the appendix."
>
> - Thank you for your valuable suggestions. We’ve followed the reviewer’s comments and added the details of the stereotype dataset (e.g., the selected stereotype groups and stereotype topics, and some examples of our user and system prompts) in the revision Section 3; we have also added illustrations to explain Fig 2 (i.e., the detailed meaning of the numbers in the cells and some findings). We have highlighted our revision in blue. Please let us know if there are further suggestions. Thank you!
>
> 2. "Out-of-distribution robustness should be illustrated in what perspective are we evaluating, that was not clear to me.”
>
> - Thank you for the valuable suggestion. Following the suggestion, we’ve added more explanations about what perspectives we aim to evaluate in the “goal” part in the revision Section 5. Specifically, For out-of-distribution robustness, we mainly focus on evaluating the robustness of GPT models under OOD instances that significantly deviate from the distribution of training or in-context demonstrations.
>
> - In particular, our study focuses on three perspectives: 1) OOD language style: evaluation of inputs with uncommon text styles (e.g., Bible style) that may fall outside the training or instruction tuning distribution, with the goal of assessing the robustness of the model when the input style is unusual. 2) OOD knowledge: evaluation of questions that can only be answered with knowledge after the training data was collected, which we aim to investigate the trustworthiness of the model’s responses when the queries are out of scope. 3) OOD in-context demonstrations: evaluation of how in-context demonstrations that are on purpose drawn from different distributions or domains from the test inputs can affect the final performance of GPT models.
>
> - We thank the reviewer for the constructive feedback, and we will leverage the additional page in the revision to add all the details and related discussions following the suggestions.

---

> ### Author Response · Authors · 2023-08-22
> **Thank you for your valuable comments**
>
> 3. “Was it expensive to run all these prompts on GPT-4? Is a normal user or a student able to run all these prompts? If you can mention this in the paper, that would be really appreciated.”
>
> - Thanks for your insightful questions! We have added details about computation and query costs in Appendix K in the revision following the suggestions. We also put a detailed analysis of costs below.
>
> - Table 1 summarizes the 1) total number of prompts, 2) total number of prompt tokens, 3) total number of completion tokens, and 4) total run costs for each trustworthiness perspective and each GPT model.
>
> *Table 1. Total computing costs of evaluation on different trustworthiness perspectives on GPT models*
> | Perspectives                                 | Models  | #/ Prompts | #/ Prompt Tokens | #/ Completion Tokens | Total Cost ($) |
> | -------------------------------------------- | ------- | ---------- | ---------------- | -------------------- | -------------- |
> | Toxicity                                     | GPT-3.5 | 49,200      | 10,966,554       | 15,796,800           | 78.14          |
> | Toxicity                                     | GPT-4   | 49,200      | 10,966,554       | 15,796,800           | 2158.97        |
> | Stereotype                                   | GPT-3.5 | 3,456       | 766,296          | 12,960,000           | 27.46          |
> | Stereotype                                   | GPT-4   | 3,456       | 766,296          | 12,960,000           | 800.58         |
> | Adversarial Robustness                       | GPT-3.5 | 42,755      | 3,596,216        | 684,080              | 9.30            |
> | Adversarial Robustness                       | GPT-4   | 42,755      | 3,596,216        | 684,080              | 162.23         |
> | OOD                                          | GPT-3.5 | 47,079      | 13,879,675       | 470,790              | 28.70           |
> | OOD                                          | GPT-4   | 47,079      | 13,879,675       | 470,790              | 444.64         |
> | Robustness against adversarial demonstration | GPT-3.5 | 233,100     | 152,882,443      | 322,259              | 306.41         |
> | Robustness against adversarial demonstration | GPT-4   | 233,100     | 144,558,043      | 256,140              | 4352.11        |
> | Privacy                                      | GPT-3.5 | 106,150    | 6,363,542        | 2,408,800            | 17.54          |
> | Privacy                                      | GPT-4   | 106,150    | 6,363,542        | 2,408,800            | 335.43         |
> | Machine Ethics                               | GPT-3.5 | 21,869     | 6,796,656        | 373,380              | 15.31          |
> | Machine Ethics                               | GPT-4   | 21,869     | 6,796,656        | 373,380              | 242.29         |
> | Fairness                                     | GPT-3.5 | 32,400     | 16,798,525       | 180,000              | 34.00             |
> | Fairness                                     | GPT-4   | 32,400     | 16,798,525       | 180,000              | 503.35         |
>
> - In addition, we further break down our evaluation for each perspective into detailed scenarios, and Table 46-53 in Appendix K show the similar computing and query costs for these detailed scenarios under different trustworthiness perspectives. Similarly, we report the 1) number of prompts for each scenario and GPT models, 2) number of tokens of the prompts, 3) number of completion tokens that answer the prompts, 4) single run cost of answering the prompts, 5) number of the run repetitions, and 6) total run cost. These tables also allow users to flexibly determine whether they want to run some subsets of the scenarios of each perspective based on their available resources.

---

> > ### Comment · Reviewer_JNyJ · 2023-08-30
> > **Useful additions**
> >
> > I see that the authors added important illustrations which is useful for all readers to understand the cost and the impact.
> > I will keep my score, this is a useful paper and I would like to see it accepted

---

### Author Response · Authors · 2023-08-22
**General Response**

We thank all the reviewers for their comments and valuable feedback. We have made the following major updates following the reviews to further improve our work.

1. Following the suggestions from Reviewer JNyJ and Reviewer UhDT, we add more details about the experimental setups of our evaluations in Section 3 and Section 5, such as the details of the stereotype dataset and the evaluation aspects of out-of-distribution robustness.

2. Following the suggestions from Reviewer imyr and Reviewer qCen, we add Section 10 in the main text, which incorporates the potential future directions to safeguard LLMs, discussing the possible strategies to address the identified vulnerabilities.

3. Following the suggestion from Reviewer qCen, we add Section 11 Related work in the main text, which provide discussions about existing benchmarks, prompt injection strategies, and regulations for the safety of AI systems.

4. Following the suggestions from Reviewer JNyJ and Reviewer UhDT, we add Appendix K, which provides the computational and query cost for all trustworthiness perspectives and specifies the 1) total number of prompts, 2) total number of prompt tokens, 3) total number of completion tokens, and 4) total run costs for each trustworthiness perspective. In addition, we further break down our evaluation for each perspective into detailed scenarios and report the computation and query costs for each scenario for different perspectives.

5. Following the suggestions from Reviewer UhDT, Reviewer j1j9, Reviewer qCen, we add Appendix L, which provides evaluations from all the trustworthiness perspectives on the top open-source LLMs in the Open LLM leaderboard, including Llama-v2-7B-Chat, Vicuna-7B, Alpaca-7B, MPT-7B, Falcon-7B, RedPajama-INCITE-7B-Instruct.

6. Following the suggestions from Reviewer imyr, Reviewer qCen, and Reviewer ADRf, we add discussions about the potential misuse and negative impacts of our datasets and discuss why it is important to publish these evaluations in Appendix N.

7. Following the suggestion from Reviewer qCen, we add discussions about DecodingTrust and existing AI regulations in Appendix Q.

All updates are highlighted in blue in our revision. If the manuscript is accepted, all contents in blue in the main text will remain given the extra page limit for the camera-ready version.

---

### Decision · Program_Chairs · 2023-09-22

**Decision:**

Accept (Oral)

**Comment:**

The paper presents an intensive and extensive study of trustworthiness in GPT models. The paper discusses different evaluation perspectives of trustworthiness using different benchmarks.

All reviewers have positive opinions about the paper.
There are minor concerns authors must take into account for the final version.